# Nanoparticle-Based Nitric Oxide Donors: Exploring Their Antimicrobial and Anti-Biofilm Capabilities

**DOI:** 10.3390/antibiotics13111047

**Published:** 2024-11-05

**Authors:** Gonzalo Tortella Fuentes, Paola Fincheira, Olga Rubilar, Sebastian Leiva, Ivette Fernandez, Mauricio Schoebitz, Milena T. Pelegrino, André Paganotti, Roberta Albino dos Reis, Amedea B. Seabra

**Affiliations:** 1Centro de Excelencia en Investigación Biotecnológica Aplicada al Medio Ambiente-CIBAMA, Facultad de Ingeniería y Ciencias, Universidad de La Frontera, Temuco 4811230, Chile; olga.rubilar@ufrontera.cl (O.R.); s.leiva04@ufromail.cl (S.L.); i.fernandez07@ufromail.cl (I.F.); 2Departamento de Ingeniería Química, Facultad de Ingeniería y Ciencias, Universidad de La Frontera, Temuco 4811230, Chile; 3Programa de Doctorado en Ciencias de Recursos Naturales, Universidad de La Frontera, Temuco 4811230, Chile; 4Departamento de Suelos y Recursos Naturales, Facultad de Agronomía, Campus Concepción, Universidad de Concepción, Casilla 160-C, Concepción 4030000, Chile; mschoebitz@udec.cl; 5Center of Biotechnology, Universidad de Concepción, Barrio Universitario s/n, Concepción 4030000, Chile; 6MirScience Therapeutics, São Paulo 04020-040, SP, Brazil; 7Departamento de Farmácia, Universidade Federal de São Paulo, Diadema 09972-270, SP, Brazil; 8Centro de Ciências Naturais e Humanas, Universidade Federal do ABC, Santo André 09606-045, SP, Brazil; roberta.reis@ufabc.edu.br (R.A.d.R.); amedea.seabra@ufabc.edu.br (A.B.S.)

**Keywords:** nitric oxide, nanoparticles, NO-donors, antimicrobials, anti-biofilms

## Abstract

**Background:** Nitric oxide (NO) is an antimicrobial and anti-biofilm agent with significant potential for combating biofilm-associated infections and antibiotic resistance. However, owing to its high reactivity due to the possession of a free radical and short half-life (1–5 s), the practical application of NO in clinical settings is challenging. **Objectives**: This review explores the development of NO-releasing nanoparticles that provide a controlled, targeted delivery system for NO, enhancing its antimicrobial efficacy while minimizing toxicity. The review discusses various NO donors, nanoparticle platforms, and how NO disrupts biofilm formation and eradicates pathogens. Additionally, we examine the highly encouraging and inspiring results of NO-releasing nanoparticles against multidrug-resistant strains and their applications in medical and environmental contexts. This review highlights the promising role of NO-based nanotechnologies in overcoming the challenges posed by increasing antibiotic resistance and biofilm-associated infections. **Conclusions:** Although NO donors and nanoparticle delivery systems show great potential for antimicrobial and anti-biofilm uses, addressing challenges related to controlled release, toxicity, biofilm penetration, resistance, and clinical application is crucial.

## 1. Introduction

Nitric oxide (NO) is a small, highly reactive, gaseous molecule that plays a crucial role in various physiological processes in humans, including vasodilation [1], neurotransmission [2], and immune response [3]. However, NO also has an essential function as a signaling molecule with varied physiological functions in plants [4] and plant–microorganism interactions [5]. NO was discovered as a signaling molecule in the late 20th century, and it has since been significantly studied for its diverse applications in medicine [6] and, more recently, in agriculture [7]. In medicine, NO has been shown to regulate blood pressure, modulate immune responses, and facilitate wound healing, making it a focal point of therapeutic research [8]. Additionally, NO plays a crucial role in neurotransmission, vasodilation, and inflammation control, contributing to its potential applications in treating cardiovascular diseases, infections, and certain cancers [6,8]. Beyond these effects, NO has demonstrated antioxidant properties, which help mitigate oxidative stress, and has been explored as a neuroprotective agent in conditions like stroke [9] and neurodegenerative disorders [10]. The NO molecule’s broad range of physiological roles and therapeutic potential highlights its significance in acute treatments and chronic disease management, making it an essential focus in biomedical research.

On the other hand, in addition to NO, the production of other reactive nitrogen species (RNS) can contribute to cellular toxicity. For example, when NO reacts with superoxide anion (O_2_•^−^), it forms peroxynitrite (ONOO^−^), a highly reactive and damaging compound that can induce oxidative and nitrosative stress [11]. This stress leads to various forms of cellular damage, including lipid peroxidation, protein nitration, and DNA modifications, all of which play a role in the pathophysiology of human diseases. The interplay between NO and other toxic nitrogen compounds underscores the need for controlled delivery of NO to minimize undesired adverse effects while maximizing its therapeutic advantages.

This attractive small molecule exerts its antimicrobial properties in two ways. When NO is present at low concentrations, it acts as a biological signal that increases or stimulates immune cell activity. On the other hand, if NO is present at high concentrations, it binds to DNA, lipids, and proteins, exerting its effects by inhibiting or killing pathogenic bacteria [9], which opens new avenues for combating infections, particularly in a time marked by rising antibiotic resistance.

The increasing occurrence of antibiotic-resistant bacteria and the persistent challenge of biofilm-associated infections require the exploration of novel, effective antimicrobial strategies. Among these, two methods have gained significant interest due to their broad-spectrum antimicrobial properties and role in biofilm disruption. The first is using metal or polymeric nanoparticles, which have shown tremendous antimicrobial potential against many pathogens, including bacteria, fungi, and viruses [12,13,14,15,16,17]. Secondly, NO has gained special attention due to its antimicrobial activity against many pathogens, such as *Pseudomonas aeruginosa*, *Escherichia coli*, *Staphylococcus aureus*, *Staphylococcus epidermidis*, *Candida albicans*, *Bacillus cereus*, and methicillin-resistant *S. aureus,* among others [18]. However, the practical application of NO in clinical settings has been limited by its high reactivity and short half-life. For example, NO’s reactivity is highlighted by its rapid interaction with superoxide (O_2_•^−^), which is extremely fast and first-order concerning [O_2_•^−^]. NO has a short half-life of 1–5 s, depending on local conditions, contributing to its limited stability in biological systems [19].

Over the last few years, researchers have developed various NO donors and delivery systems to overcome these limitations. Nanoparticles have emerged as a promising vehicle for NO delivery, providing means to stabilize and control the release of this reactive molecule [18,20]. Nanoparticles can be engineered to encapsulate NO donors or to generate NO in situ, allowing for targeted and sustained delivery [21]. This targeted delivery enhances the antimicrobial efficacy of NO and minimizes its potential cytotoxicity to host tissues.

The antimicrobial potential of NO-releasing nanoparticles is further amplified by their ability to penetrate and disrupt biofilms [22,23]. Biofilms are structured communities of microorganisms embedded in a protective extracellular matrix that are notoriously resistant to conventional antibiotics and immune responses. The unique physicochemical properties of nanoparticles enable them to traverse the biofilm matrix and deliver NO directly to the microbial cells. This direct delivery can lead to biofilm dispersal, enhanced susceptibility of the microorganisms to antimicrobial agents, and, ultimately, more effective infection control. Effective *P. aeruginosa*, *E. coli*, *S. aureus*, *S. epidermidis*, and *C. albicans* biofilms were controlled and killed (≥99.999% killing) using NO-releasing silica nanoparticles [22].

In addition to their antimicrobial and anti-biofilm capabilities, NO-releasing nanoparticles offer advantages such as tunable release kinetics, biocompatibility, and the potential for functionalization with targeting ligands [24,25]. These properties can be tailored to specific clinical needs, making NO-releasing nanoparticles versatile tools for the fight against infectious diseases. Moreover, developing such nanoparticles involves a multidisciplinary approach encompassing materials science, chemistry, microbiology, and nanomedicine.

Exploring NO donors and nanoparticle delivery represents a frontier in antimicrobial research with significant clinical implications. Researchers aim to develop effective treatments for antibiotic-resistant infections and biofilm-associated conditions by harnessing the unique properties of NO and designing innovative nanoparticles. This introduction sets the stage for a comprehensive examination of the current advancements, challenges, and future directions of NO-releasing nanoparticles for antimicrobial and anti-biofilm applications. This review will delve into these topics, providing an in-depth analysis of NO-releasing nanoparticle strategies, mechanisms, and potential clinical applications.

## 2. Nitric Oxide Donors

From the earliest publications describing NO’s pharmacology effects [26,27] in the 1970s and 1980s to 2024, more than 340,000 articles were published and 59,600 patents were registered to deepen the knowledge of NO’s biological range of effects, according to the Web of Science Core Collection and Derwent databases, respectively. In this context, the pharmacological effects of NO have been described as promising for addressing, mitigating, and managing a multitude of pathologies [26,27,28].

The range of physiological effects of NO is due to its vasodilation effect [26,27,28], and all applications in the cardiovascular field, including therapies and biomaterials, are due to its antimicrobial effect [29] and potential as oncotherapy [30]. However, the amount of NO released per unit of time at the target site remains a technical challenge to achieve its true potential. Thus, it is not just the concentration of the drug at the target site, as with traditional small molecule drugs, but rather the sustained rate of NO release at the target site that is crucial. Conventional drug delivery strategies are not designed to meet this requirement effectively. Therefore, two methods that can be applied separately or in combination have been the focus of significant attention: molecules capable of releasing NO donors and nanotechnology.

### 2.1. Types of Nitric Oxide Donors

The most commonly used NO donors are *S*-nitrosothiols (RSNO), *N*-diazeniumdiolate (NONOates), nitrates, furoxan, metal nitrosyl complexes, and nitrobenzenes. Figure 1 shows each type’s functional group, an example molecule, and the NO release mechanism for all classes. All these have unique characteristics that should match the desired application requirements.

Figure 2 shows the articles and patents for each NO donor type described, according to the Web of Science Core Collection and Derwent databases, respectively (research made on 24 April 2024). According to articles published, the RSNOs are the most used and metal nitrosyl complexes, according to a registered patent.

The NO donor class with the highest number of recent publications is RSNOs, which include molecules with an *S*-nitrosothiol functional group, such as *S*-nitrosoglutathione (GSNO), *S*-nitrosocysteine (CySNO), and *S*-nitroso-*N*-acetylcysteine (SNAC), as shown in Figure 1 [25,26,27].

Figure 2 demonstrates the disproportionate ratio between patents and scientific publications. This disparity is likely influenced by the commercial availability and established roles of different NO donors, such as RSNOs and nitrates, often used in various applications, including agriculture and the environment [31,32]. While RSNOs have more research articles, nitrates are more balanced in both patents and publications, reflecting their broader regulatory approval and commercialization.

The second NO class donor with the most publications is the metal nitrosyl, which includes sodium nitroprusside (SNP), as shown in Figure 1. SNP has been approved by the Food and Drug Administration (FDA) since 1974 for acute hypertension [27,28,29,30,31,32,33,34]. In addition, several clinical trials are ongoing for different indications, including acute lung injury (NCT01619280) [35], thyroidal function (NCT00636584) [36], and schizophrenia (NCT01548612) [37].

Nitrates are the third most used class of NO donors in research. They are characterized by a nitrate functional group, such as nitroglycerin, as shown in Figure 1. Nitroglycerin was approved by the FDA in 2000 for acute relief of an attack of angina pectoris secondary to coronary artery disease, and it is administered as a sublingual tablet [38]. In addition, multiple clinical trials have explored other potential cardiovascular applications of nitrates, such as hypertension, heart failure, and preventing myocardial infarction [39,40,41].

The most commonly used classes of NO donors are NONOates, furoxans, and nitrobenzenes, characterized by the [N(O)NO]^−^, furoxan, and nitrobenzenes functional groups, respectively, as shown in Figure 1 [31,33].

Overall, NO donors offer therapeutic potential to reach several indications, but their physiological effects can be further enhanced and tailored by combining them with other innovative strategies, such as nanoparticles. Recent research highlights this approach as holding considerable promise for advancing the field of NO-based therapeutics [42].

### 2.2. Mechanisms of Nitric Oxide Release

The mechanisms of NO release can be divided into the following categories: entirely spontaneous, spontaneous with modulation, and non-spontaneous. To achieve effective NO-based therapy, it is essential to control the rate of NO release over time, as it plays a crucial role in its therapeutic application. In this sense, modulating the NO release is key to practical application.

The entirely spontaneous category includes the NO donors’ release of NO in a non-controllable regimen. It includes nitrates, such as nitroglycerin and nitrobenzene. The spontaneous release with modulation category consists of NO donors that modulate the NO flow by irradiation and/or thermal stimuli, including metal nitrosyl complex, RSNOs, and NONOates. The non-spontaneous category includes furoxan, which requires a thiol attack to start the NO release (Figure 1).

Each category should be paired with the desired physiological effects to achieve maximum benefits. Observing the NO concentration over time that each category can achieve in a specific environment is relevant to matching the range necessary for the intended physiological effect.

### 2.3. Applications of Nitric Oxide Donors

The wide range of NO donor applications has been highlighted by diverse studies involving microbicidal to antitumoral effects. Cai and Webb [33] tested seven different NO donors, including one metal nitrosyl (SNP), two RSNOs, and five NONOates, against *P. aeruginosa* biofilms. In a microtiter plate-based biofilm assay, seven NO donors were tested at nine concentrations (1 µM to 500 µM) in different incubation periods (1 h to 24 h). The results showed that two NO donor treatments had optimal performances: SNP and the NONOATE Spermine S150. SNP has a half-life lower than 2 min and Spermine S150’s is 39 min. Despite the difference in half-life, the results at 250 µM after 24 h of incubation showed similar performances in dispersing *P. aeruginosa* biofilms of 63.5% and 60.0% for SNP and S-150, respectively. Interestingly, within 1.5 h, the SNP was only 29.2% in the *P. aeruginosa* biofilm’s biomass reduction. Therefore, studying the kinetics of NO release is crucial to understanding the pharmacokinetic profiles of safety and efficacy parameters that are strongly linked with dosage and dosage regimen [33]. 

Similarly, Guo et al. [29] studied four RSNO donors, GSNO, CySNO, SNAC, and 2-(2-*S*-nitroso propionamide) acetic acid (GAS), against *E. coli* and *S. aureus* using the inhibition zone method. The results indicated potent antibacterial activity in the order of SNAC > GSNO > CySNO > GAS [29].

Zhang et al. [43] synthesized a polymer based on propylene oxide with *N*-diazeniumdiolate moieties. They showed NO releases of 0.4, 0.5, and 0.6 µmol/mg of gaseous NO from 1.0 mg of the NO donor based on PEI600, PEI1800, and PEI10K. The in vitro data showed biocompatible levels, and the in vivo study showed that the groups treated with NO had higher rates of healthy tissue than infarction in rats with a model of middle cerebral artery occlusion (MCAO) [43].

Li et al. [44] studied the effect of NO donors (SNAP, SNP, and ISMN) in tumor progression in immune-competent mice. The results showed that low doses of NO donors inhibited tumor growth in two immune-competent mouse models, but not in the immune-deficient mouse model. The RNA-seq data showed an activation of genes related to CD8+T cells, and inhibiting these cells removed the therapeutic effect. Low concentrations of exogenous NO donors inhibit tumor growth in vivo by regulating T cells and macrophages. In addition, low-dose NO donors may be combined with chemotherapeutic drugs for cancer therapy [44].

Overall, NO donors have a wide range of physiological applications, and parameters such as the half-life and concentration of NO release are crucial to achieving successful application. In this context, combining NO donors and nanoparticles provides a platform for multiple applications with modulation of NO release.

## 3. Nanoparticles for Nitric Oxide Delivery

### 3.1. Types of Nanoparticles Used for Nitric Oxide Delivery

Numerous types of nanoparticles can serve as carriers for NO donors, each with unique characteristics and advantages. However, this review does not focus on chemically synthesizing these nanoparticle platforms or their detailed characterization. For readers interested in this aspect, we recommend consulting other comprehensive reviews on the topic, such as [45,46,47].

Nanoparticles have been explored for NO delivery, each with distinct characteristics and potential applications. Researchers have investigated various nanoparticles to effectively deliver NO to target bacterial biofilms, which are notoriously resistant to conventional antibiotics. As presented in Table 1, types of NO-delivery nano vehicles documented in the literature for antimicrobial and anti-biofilm formation include silica, polymeric, copolymeric [48], lipidic [49,50,51], metals-based [52,53,54,55], carbon dots [56], and graphene derivatives [57,58,59].

Polymeric nanoparticles are formed from biodegradable and biocompatible polymers such as polyglycolic acid (PGA), polylactic acid (PLA), and their copolymers (PLGA) [60,61,62,63,64,65]. These nanoparticles can be engineered to encapsulate NO donors, enabling the controlled and sustained release of NO. Their stability and easy functionalization make them ideal candidates for biomedical applications, especially for facilitating the co-delivery of gasotransmitters and antibiotics [62]. These polymeric approaches have been exhaustively studied against *S. aureus.* NO-releasing nanoparticles integrated with a chitosan hydrogel–glass composite showed enhanced antimicrobial activity and prevented biofilm formation on medical devices, such as central venous catheters [66]. Another use is PLGA-based nanoparticles loaded with isosorbide mononitrate, which provide sustained NO release and show significant antibacterial activity against *S. aureus* biofilms [60]. Co-assembled NO-releasing nanoparticles with Pluronic F127 exhibited potent antimicrobial efficacy against methicillin-resistant *S. aureus* strains, offering the potential for a new antimicrobial recipe design [67].

**Table 1 antibiotics-13-01047-t001:** Nanoparticles for NO delivery for antimicrobial and antibiofilm activity MIC (Minimum Inhibitory Concentration). MBEC (Minimum Biofilm Eradication Concentration).

Nanoplataform	NO Donor	MIC; MBEC *	Effective Against	Ref.
Silica Nanoparticles	*N*-diazeniumdiolate (NONOate)	8 mg/mL	*P. aeruginosa; Escherichia coli*; *S. aureus*; *S.epidermidis;*	[22]
PEG- derivative	*N*-nitrosated naphthalimide	0.877 mg/mL; 0.8 m/mL	*Escherichia coli*; *S. aureus*	[23]
Liposome (egg lecithin)	isosorbide mononitrate (ISMN)	60 mg/mL	*S. aureus*	[49]
Copper-Based Metal Organic Framework	*S*-nitroso-*N*-acetyl-penicillamine (SNAP)	NO-MOF-3% **	Methicillin-resistant *S. aureus*; *Escherichia coli.*	[52]
Polydopamine-Coated Iron Oxide (PDA-IONPs)	*N*-diazeniumdiolates (NONOate)	6 and 12 × 10^−6^ mg/mL of NO	*P. aeruginosa*	[53]
Gold nanocages (AuNC@NO)	*N*-nitroso(4-mercaptomethylphenyl)-hydroxylamine (TCup)	100 p mol/L; 400 p mol/L of NO	Methicillin-resistant *S. aureus*	[54]
Carbon dot	*N*-diazeniumdiolate (NONOate)	1 mg/mL; 1 mg/mL	*P. aeruginosa*	[56]
Polyethylenimine graphene oxide (GO-PEI) nanoparticles	(GO-PEI/NO)	0.1 mg/mL; 0.1 mg/mL	Methicillin-resistant *P. aeruginosa*; Methicillin-resistant *S. aureus*.	[58]
Graphene oxide nanosheets (GO)	(GO)-*S*-nitrosothiol (GOSNO)	500 μg/mL; 500 μg/mL	Methicillin-resistant *S. aureus*; *E.coli*	[59]
Poly(lactide-co-glycolide) (PLGA)	isosorbide mononitrate (ISMN)	30 mg/mL 60 mg/mL	*S. aureus*	[60]
Chitosan- hydrogel-glass composite	NaNO_3_	5 mg/mL 5 mg/mL	Methicillin-resistant *S. aureus*	[65]
Pluronic F127	furoxan derivatives	16 ug/mL	Methicillin-resistant *S. aureus*	[66]
Gold Core@Shell Mesoporous Silica	AuNR@MSN-SNO	50 μg/mL	*S. aureus*	[68]
Silver Nanoparticles in alginate hydrogel	*S*-nitroso-mercaptosuccinic acid (MSA)	2 μg/mL	*Escherichia coli;* *S. aureus*; *Streptococcus mutans*	[69]

* For information about time dependence, please check the reference source. ** Relative value for scaffold used in the study in reference [52].

Lipid-based nanoparticles, including liposomes and solid lipid nanoparticles (SLNs), have been used for NO delivery. Since liposomes are spherical vesicles composed of lipid bilayers, they can encapsulate hydrophilic and hydrophobic NO donors, modulating the interaction of the pathogen’s membrane. SLNs, on the other hand, are composed of solid lipids, offering a more stable and controlled release platform for NO [51]. For example, liposomal encapsulation of NO precursors such as isosorbide mononitrate enhances their anti-biofilm effects against *S. aureus*, showing potential as an effective topical agent for clinical use [49].

Mesoporous silica nanoparticles (MSNs) feature a highly ordered porous structure, providing a high surface area for loading NO donors. The tunable pore size and surface chemistry of MSNs enable controlled and targeted NO delivery, making them suitable for therapeutic applications. NO-releasing silica nanoparticles have shown high efficacy in killing biofilm-based microbial cells, including *P. aeruginosa, E. coli, S. aureus, S. epidermidis*, and *C. albicans*, with the most remarkable efficacy against gram-negative bacteria [22,68]. Smaller silica nanoparticles (14 nm) and those with higher aspect ratios are more effective in delivering NO and eradicating biofilms than larger or spherical particles [68]. In a combination of MSN and metal nanoparticles, gold Core@Shell mesoporous silica nanoparticles combine photothermal therapy with NO release and antibiotic delivery, significantly reducing *S. aureus* biofilm integrity and bacterial viability upon near-infrared irradiation [69].

Metallic nanoparticles, particularly those made from gold and silver, possess unique optical and electronic properties, facilitating NO delivery [54,69,70]. One example is polydopamine-coated iron oxide nanoparticles (PDA-IONPs), which release NO and show significant bactericidal activity and biofilm dispersal, particularly against *P. aeruginosa* [53]. Copper-based metal–organic frameworks in polymeric substrates with embedded NO donors effectively reduce bacterial growth, preventing biofilm formation on medical devices by planktonic methicillin-resistant *S. aureus*, with similar results observed for *Escherichia coli* [52].

Finally, carbon-based nanoparticles have been studied for NO delivery antimicrobial action. Utilizing chitosan-graft-poly(amidoamine) dendrimer (CPA), fluorescent carbon dots were loaded with NO via the formation of *N*-diazeniumdiolate (NONOate), yielding CPA-CDs/NONOate that exhibited 3.5 times the NO content compared to the CPA copolymer and were tested successfully against gram-negative *P. aeruginosa* [56]. Another carbon approach is graphene oxide (GO) with NO donors against *E. coli*, MRSA, and MRPA. Retained biocompatibility and the differences in surface properties compared to purine GO imparted significantly improved antibacterial and antibiofilm effects [58,59].

### 3.2. Strategies for Nitric Oxide Loading onto Nanoparticles

Due to its free radical and gaseous molecule nature, NO has a short half-life of only 1–5 s. The effectiveness of NO in biological applications hinges on the duration of its exposure and its kinetic behavior at the target site. To improve NO’s stability and enable its application, NO donors have been modified, encapsulated, adsorbed, and embedded in different matrices [71]. In other words, nanoparticles’ NO donor type and NO loading method are congruent steps. Several strategies have been developed to achieve high stability, and are explicitly designed for the nanoparticle type and the intended application.

One strategy is encapsulation. It involves physically entrapping NO donors within the nanoparticle matrix, as shown in Figure 3. This method is commonly used with polymeric and lipid-based nanoparticles, where the NO donor is mixed with the nanoparticle precursors during synthesis. In addition, carbon matrices can be modified, leading to a NO-carbon derivative that excludes the need to input another molecule into the system [56,69]. Another strategy is surface functionalization, which involves chemically attaching NO donors to the surface of nanoparticles as the in situ formation of *N*-diazeniumdiolates [72]. This approach benefits metallic nanoparticles, where NO donors can be conjugated to the nanoparticle surface via covalent bonds [73]. Adsorption entails the physical adsorption of NO donors onto the surfaces of nanoparticles. This method is often employed with MSNs, where the high surface area and porosity facilitate the adsorption of NO donors [74].

### 3.3. Controlled Release of Nitric Oxide from Nanoparticles

Controlled release of NO is crucial for maximizing its therapeutic efficiency while minimizing potential side effects. Several mechanisms have been developed to release controlled NO from nanoparticles [59,67]. Diffusion-controlled release relies on the gradual diffusion of NO donors from the nanoparticle matrix into the surrounding environment. This mechanism is often observed in polymeric and lipid-based nanoparticles, where the release rate can be modulated by altering the polymer composition or lipid content. Nanoparticles can achieve controlled drug release primarily driven by diffusion, which can be fine-tuned by creating a radial drug concentration gradient within the nanoparticle [75]. Encapsulation techniques, such as microencapsulation of nanoparticles into microparticles, can significantly reduce the initial burst release by creating a double polymeric wall that slows down drug diffusion [76]. Coating nanoparticles with materials like chitosan can reduce the initial burst without affecting the overall release profile [65]. NO, when adsorbed on the porous-like nanoparticle as silica, zeolite, or a metal–organic framework [77], can be controlled by the desorption kinetics, which can be modulated by adjusting the surface properties of the nanoparticles [78,79].

Besides the diffusion and the desorption being modulators of NO delivery, all the most recent developments have used a stimuli-responsive release mechanism of the gasotransmitter. Involving external stimuli to trigger NO release from nanoparticles presents a state-of-the-art nanomedicine design, especially if the trigger is microenvironment modulation, such as pH and enzymes [80]. Beyond the microenvironment, external stimuli can be produced by light, strain, temperature, ultrasound, and magnetism [81,82]. For example, light-responsive nanoparticles can release NO upon exposure to specific wavelengths of light, allowing for spatiotemporal control over NO delivery. Lipid and polymer nanoparticles improve the delivery of antibiotics to bacterial cells within biofilms, thereby enhancing antimicrobial efficacy. Nanoparticles enabling photo-controlled NO release offer targeted and visible NO delivery. This method has shown efficient antibacterial and anti-biofilm effects, providing a controllable approach to NO therapy [23].

In conclusion, the development of nanoparticles for NO delivery holds great promise for a wide range of biomedical applications. Controlled and efficient NO delivery can be achieved by carefully selecting the type of nanoparticle and employing appropriate loading and release strategies, thereby enhancing its therapeutic potential.

## 4. Antimicrobial and Anti-Biofilm Potential

### 4.1. Antimicrobial Activity of NO-Releasing Nanoparticles

The antimicrobial activity of NO has been extensively studied, demonstrating effectiveness against various pathogens, including bacteria (gram + and -), fungi, and viruses [13,14,15,16,17]. NO exerts its antimicrobial effects through multiple mechanisms, making it a potent and versatile agent. However, NO’s inherent instability and high reactivity have traditionally limited its practical application. The advent of NO-releasing nanoparticles has provided a novel solution, enabling NO-controlled and targeted delivery, thereby enhancing its antimicrobial potential.

The antimicrobial effects of NO are mainly mediated in several ways, such as the induction of oxidative stress, disruption of cellular respiration, or interference with DNA synthesis [83,84]. NO can react with molecular oxygen and superoxide (O_2_•^−^) to form reactive nitrogen species such as peroxynitrite or dinitrogen trioxide. These RNS induce oxidative stress in microbial cells by damaging cellular components, including lipids, proteins, and DNA. This oxidative damage disrupts vital cellular functions and leads to cell death [85,86]. Furthermore, NO can inhibit essential enzymes involved in the respiratory chain, including cytochrome oxidase and other proteins containing iron–sulfur clusters. For example, due to the NO affinity of protein-bound iron, NO can inhibit key enzymes that contain iron in their catalytic centers [86]. This inhibition disrupts the electron transport chain, decreasing ATP production and energy depletion in microbial cells. NO and its derivatives can also cause nitration and deamination of DNA bases, leading to mutations and inhibition of DNA replication. This interference hampers microbial proliferation and influences the antimicrobial effect of NO.

Nanoparticles offer distinct advantages as carriers for NO donors. These include targeted delivery, controlled release, enhanced stability, and reduced toxicity. Nanoparticles can be engineered to target specific sites of infections. Functionalization with ligands or antibodies allows for the selective delivery of NO to infected tissues, reducing off-target effects and enhancing local antimicrobial activity. Estes et al. [87] reported that a combined solution of cerium oxide nanoparticles (CNP) and a NO donor, *S*-nitroso-*N*-acetylpenicillamine (SNAP), exhibited high antimicrobial activity against *S. aureus, E. coli,* and *C. albicans.* Their study demonstrated that an equimolar solution of 3 mmol/L SNAP and CNP was more effective against these microbes than higher concentrations of each agent used individually, indicating the potential for this combination to be developed into broad-spectrum antimicrobial coatings, particularly for use in biomedical devices. In other work, Yu et al. [88] reported that the integration of photothermal and NO-releasing properties in a single nanocomposite, Fe_3_O_4_@PDA@PAMAM@NONOate, led to efficient bactericidal effects against both gram-negative and gram-positive bacteria through membrane damage and intracellular component leakage under laser irradiation. Duan et al. [89] reported the development of a micellar nanoparticle platform that simultaneously delivers NO and formaldehyde (FA) under visible light irradiation, demonstrating combinatorial antibacterial effects against both gram-negative and gram-positive bacteria with low toxicity to mammalian cells.

However, the release kinetics of NO can be precisely controlled by the design of the nanoparticle carrier, ensuring a sustained release of NO and maintaining effective antimicrobial concentrations over extended periods. In this sense, several nanoparticles or nanomaterials have been designed for NO delivery. Hetrick et al. [90] demonstrated that NO-releasing silica nanoparticles show enhanced antibacterial efficacy against *P. aeruginosa* compared to small-molecule NO donors such as 1-[2-(carboxylate)pyrrolidin-1-yl]diazen-1-ium-1,2-diolate (PROLI/NO), with reduced cytotoxicity to healthy cells, supporting the advantage of nanoparticle-based NO delivery for antimicrobial applications. Encapsulation of NO donors within nanoparticles protects them from premature degradation and preserves their activity until they reach the target site, enhancing the stability and bioavailability of NO. By localizing NO delivery to the site of infection, nanoparticles minimize systemic exposure and potential cytotoxicity to host tissues, improving the safety profile of NO-based therapies.

NO-releasing nanoparticles have demonstrated efficacy against antibiotic-resistant strains of bacteria, including methicillin-resistant *S. aureus* (MRSA) and multidrug-resistant *P. aeruginosa*. These pathogens pose significant clinical challenges due to their resistance to conventional antibiotics. In this sense, Ref. [91] reported that applying sustained NO-release nanoparticles against methicillin-resistant *S. aureus* skin infections showed significant antimicrobial activity. These authors reported that these nanoparticles accelerated wound closure in a murine wound model, reduced bacterial burden, decreased suppurative inflammation, and minimized collagen degradation [91]. Liu et al. [67] explored a novel NO class-releasing nanoparticles using a co-assembly method with Pluronic F127. The study showed that these nanoparticles had potent antimicrobial effects against methicillin-resistant *S. aureus* (MRSA) strains. The co-assembled nanoparticles displayed fourfold improved antimicrobial effectiveness compared to their self-assembled versions. Experiments using 5(6)-carboxylfluorescein (CF) leakage demonstrated stronger interactions with lipid bilayers, further confirmed by high-resolution optical and scanning electron microscopy, revealing significant plasma membrane damage in bacteria [67]. The unique mechanisms of action of NO, combined with the targeted delivery afforded by nanoparticles, make NO-releasing nanoparticles a promising solution for overcoming antibiotic resistance.

In conclusion, NO-releasing nanoparticles represent a novel and effective strategy for enhancing the antimicrobial potential of NO. By providing targeted, controlled, and stable delivery of NO, these nanoparticles offer a versatile approach to combating a wide range of pathogens, including antibiotic-resistant strains. Continued research and development in this field promise to improve clinical outcomes and address the growing challenge of antimicrobial resistance.

### 4.2. Anti-Biofilm Properties of NO

Biofilms are complex bacterial communities embedded in a self-produced extracellular polymeric substance (EPS), which protects against environmental stresses, including antimicrobial agents and the host immune system [92]. Forming biofilm is a significant factor in the persistence and chronicity of infections, particularly in medical devices and chronic wounds [93]. In this sense, the use and application of NO have shown significant potential in combating biofilms due to their unique properties, as reported by [48]. NO’s small size and diffusibility allow it to penetrate the biofilm matrix effectively. Unlike many conventional antibiotics, which struggle to penetrate the dense extracellular matrix, NO can diffuse rapidly through the biofilm layers, reaching the embedded pathogens [48]. Once within the biofilm, NO exerts its antimicrobial effects through oxidative stress and enzymatic inhibition, disrupting the biofilm structure [22]. Hetrick et al. [22] reported that NO-releasing silica nanoparticles effectively killed biofilm-forming cells of *S. aureus*, *C. albicans*, *E. coli*, *S. epidermidis*, and *P. aeruginosa*. The treatment eradicated ≥ 99% of biofilm cells, with the highest efficacy against gram-negative *P. aeruginosa* and *E. coli*. Cytotoxicity tests showed the nanoparticles were less harmful to fibroblasts than commonly used antiseptics like chlorhexidine, highlighting their potential for antimicrobial delivery to biofilms. da Silva Filho [94] investigated silica nanoparticles containing nitroprusside (MPSi-NP) as NO donors against methicillin-sensitive and methicillin-resistant *Staphylococcus* strains. MPSi-NP exhibited 63% NO release in 24 h, compared to 18% for free nitroprusside. Although MPSi-NP showed moderate biofilm inhibition, it significantly reduced viable bacterial cells over 600-fold for *S. aureus* and *S. epidermidis*. Additionally, MPSi-NP combined with ampicillin reduced the MIC for resistant *S. aureus* (twofold) and *S. epidermidis* (fourfold). A carbopol-based gel with MPSi-NP also showed inhibition zones, suggesting potential for topical treatment of resistant bacterial infections.

NO can disrupt the biofilm matrix by breaking down the EPS that holds the biofilm together. This breakdown is facilitated by RNS generated by NO, which can degrade proteins, polysaccharides, and nucleic acids within the EPS. The EPS degradation weakens the biofilm’s structural integrity, making it more susceptible to antimicrobial agents and the host immune response. Cai and Webb [33] evaluated the effectiveness of Spermine NONOate (S150) as an NO donor for dispersing *P. aeruginosa* [33] biofilms, particularly those associated with cystic fibrosis infections. Their results showed that S150 reduced more than 60% of the biofilm biomass within just 2 h, performing significantly better than other NO donors like sodium nitroprusside (SNP), which required light activation and a longer time to achieve similar results.

Furthermore, S150 was tested on 13 CF clinical isolates of *P. aeruginosa*, with most biofilms showing substantial dispersal at a concentration of 250 μmol/L. Notably, S150 released NO spontaneously and did not produce cyanide, making it a promising alternative to SNP for future clinical applications. Ma et al. [95] developed a surface charge-adaptable NO nanogenerator (PDG@Au–NO/PBAM) that enhances photothermal therapy for biofilm eradication. This nanogenerator changes from negatively charged physiological environments to positively charged in acidic biofilm microenvironments, promoting deep biofilm penetration. It releases NO to disrupt biofilms, sensitizing bacteria for further treatment. Under near-infrared irradiation, the combined effects of hyperthermia and NO release effectively eradicated drug-resistant *MRSA* biofilms in vitro and in vivo, outperforming vancomycin. This strategy shows promise for overcoming biofilm barriers and combating drug-resistant bacteria.

One of the key anti-biofilm properties of NO is its ability to induce biofilm dispersal [96]. NO signaling can trigger the dispersion of microbial cells from the biofilm matrix. This process involves the downregulation of biofilm-promoting genes and the upregulation of motility and planktonic growth genes. This dispersal increases the susceptibility of the bacteria to antimicrobial treatments and immune clearance. In this sense, Ref. [97] studied how NO triggers biofilm dispersal in *P. aeruginosa*. These authors reported that low NO concentrations activated phosphodiesterases, reducing intracellular c-di-GMP (secondary messenger) levels and promoting biofilm dispersal. Moreover, NO also upregulated motility and energy metabolism genes while downregulating adhesins and virulence factors. Mutagenesis studies identified the chemotaxis transducer BdlA as a critical factor in the NO-induced biofilm dispersal response [97]. Several studies have shown that NO can generate biofilms in various bacterial species, including *P. aeruginosa* and *S. aureus*. The dispersal effect is dose-dependent, with low concentrations of NO promoting biofilm formation and higher concentrations inducing dispersal. This dual effect underscores the importance of controlled NO delivery for achieving effective biofilm disruption.

Sustained NO-releasing nanoparticles have shown significant potential in preventing and disrupting biofilm formation. Mihu et al. [66] showed that NO-releasing nanoparticles disrupted *S. aureus* adhesion and inhibited biofilm formation in a rat model of central venous catheter infection. Confocal and scanning electron microscopy confirmed reduced biofilm thickness and bacterial numbers, both in vitro and in vivo. In addition to disrupting existing biofilms, NO can prevent the initial formation of biofilms by interfering with the adhesion of bacteria to surfaces, a critical first step in biofilm development. By inhibiting adhesion, NO reduces the likelihood of biofilm establishment and subsequent infection, highlighting its potential for use on medical devices such as CVCs. Furthermore, incorporating chitosan into the NO-np enhanced its antimicrobial properties, making it a promising prophylactic or therapeutic strategy against bacterial biofilms [66].

NO’s ability to modulate bacterial communication (quorum sensing) [98] further contributes to its anti-biofilm properties. Quorum sensing regulates the expression of biofilm-related genes, and NO can disrupt these signaling pathways, inhibiting biofilm maturation and maintenance [99,100]. The anti-biofilm properties of NO have significant clinical implications, particularly in managing chronic wounds, catheter-associated infections, and implant-related biofilms. NO-releasing nanoparticles can be applied to wound dressings, catheter coatings, and other medical devices to prevent biofilm formation and promote the clearance of existing biofilms. The combination of NO’s antimicrobial and anti-biofilm activities offers a multifaceted approach to infection control. By targeting planktonic and biofilm-associated bacteria, NO-releasing nanoparticles can enhance the efficacy of existing antimicrobial therapies and reduce the incidence of chronic infections.

## 5. Applications of Nitric Oxide Donors and Nanoparticles

As the discovery of the biological role of the NO molecule revolutionized the biochemical drug world, the necessity to manipulate its quantities in controlled sites of organisms became clear. One thing that would prove essential to this is enhancing the half-life stability of NO in-vivo. Due to its reactivity, this molecule has a very short half-life as a free radical. To contradict this, the first technological wave to improve was the NO donors that released NO in a controlled manner, as discussed in Section 3. After this wave, mainly in the last two decades, a new wave of strategy emerged with the intersection between nanoparticles’ revolution and nitric oxide’s importance, as discussed in Section 3.

Chronologically speaking, although the first applications of NO were described in the 1990s with famous patents, the technological transfer would only occur in the 2000s. The NO donor and nanoparticle delivery systems would follow in the next two decades, creating new applications. This transfer from university to industry involves moving research findings from academic settings to commercial applications and, consequently, the university filing for intellectual property protection. The sector then develops new patents, so observing a delay between scientific production and commercial application is normal, as seen in Figure 4. These figures show the research papers in Scopus and the intellectual property documents in Lens.org.

As shown in Figure 4, both NO donors and nanoparticles are very promising as product technology, doubling their percentage share in overall NO when university is compared with industry. The ownership of the patents can also evaluate the technological transfer. As mentioned, new areas start with the university patenting new technologies. This factor is presented in Figure 4a, especially NO nanoparticles, a new field with more than one-third of its patents owned by universities.

Another apparent trend evidenced by Figure 5a is that most of the top owners besides universities are related to medical applications. This relation is further established by organizing the patents by the top Cooperative Patent Classification (CPC) codes, as shown in Figure 5b. These codes are an extension of the IPC and help to find regulatory trends. As can be seen, all top 15 codes begin with A61P, with A being “human necessities”, 61 referring to “medical or veterinary science hygiene”, and P denoting “specific therapeutic activity of chemical compounds or medicinal preparations”, showcasing the important role of such materials in modern medicine. In addition, the most prevalent code used in the patents is A61P35/00-Antineoplastic agents, showcasing a vital role in fighting cancer, a multi-trillion-dollar market [101].

With the aging of and increase in the world population, products that address the population’s health and how to sustain this growing population will be a must in the future. NO and its donors and nanoparticles show promising medical applications, as mentioned. Still, a new research focus emerges in using these in different organisms, such as plants. The environmental applications of NO delivery systems are a new frontier promoting a new approach to maintaining the ever-increasing food demands of society, promoting increased crop yields, and, most importantly, considering climate change’s resistance to biotic and abiotic factors. Therefore, this topic will explore both medical and environmental applications.

### 5.1. Medical Applications

As discussed, NO is linked to medical applications because of its role in the human body, and this is further evidenced by the first two patents to be granted for NO donors, “Devices for treating pulmonary vasoconstriction and asthma” and “Linsidomin for Treating Erectile Dysfunctions”, both published in 1992 [102,103]. However, over three decades have elapsed, and NO has proven its capabilities in the medical field in terms of its pulmonary and vascular importance.

Considering its pulmonary role, for example, NO has shown potential to treat pulmonary hypertension [104] and threats such as coronavirus infections, including COVID-19 [105,106]. Recent studies highlight the NO’s antiviral activity, its benefits in treating respiratory viral diseases, and the promising use of NO donors with nanoparticles to enhance localized delivery. Regarding its vascular importance, its action in endothelial cells can be used to overcome vascular diseases [107,108], implants and grafts [109,110], and health failure [111], and can even present both therapeutical and diagnostic power [112,113].

Since these initial discoveries related to the vascular importance of NO, new studies have shown the importance of body-wide endogenous NO. NO delivery systems have found new horizons to act. As such, the NO now has an essential role as a new anticancer drug, as evidenced by patent deposits in antineoplastic agents. It was shown in [44] that, in the last year, low-dose NO donors (specifically *S-*Nitroso-*N*-acetyl-DL-penicillamine) activated T cells and regulated macrophages with antitumor properties in the tumor microenvironment, reducing the expression of anti-inflammatory cytokines. It showed a synergetic relationship between the donor and cisplatin, indicating that NO donors could be a promising approach for enhancing cancer immunotherapy.

This synergetic effect was also found to help fight another global health threat, antimicrobial resistance. Rouillard et al. showcase this effect against ESKAPE organisms and films [114]. The changes that NO promotes in the cell wall, promoting higher permeability to traditional antibiotics, could present a fast answer to this glooming resistance problem. Due to the number of approved FDA NO donors [44], traditional antibiotics could represent a fast deployment strategy against AMR [48].

### 5.2. Environmental Applications

Nitrogen acts as a building block for both animals and plants. Its species are almost omnipresent in living organisms, as NO plays an essential role in animal health and nitrogen is a vital source element for plants. Therefore, it is natural to investigate the influence of NO and its delivery systems in plants. NO functions as an essential signaling molecule in plants. Due to its ability to traverse cell membranes, it enhances communication between cells. This crossing of cell membranes by NO allows it to play a critical role in regulating oxidative stress, antioxidant defense mechanisms, metal transport, ion balance, and even transcriptional factors within the plant, contributing to growth, development, and stress responses [115].

Addressing this issue is crucial, as anthropogenic environmental impacts impose new challenges on our crops and threaten overall survival, demanding new strategies for increased soil contamination, sun exposure, and drought [116,117,118]. Considering, for example, contaminated soils, an example can be taken from Ahmad’s work [119]. His study examined how mercury affects soybean cultivars and found that mercury stress reduced growth and biomass yield. Simultaneously, treatment with NO donors mitigates these effects by improving growth gas exchange and reducing oxidative damage. However, regarding soybeans, NO can be used even in the soil to protect them against copper stress, as shown by [120].

Tackling the drought factor, Oliveira et al. [121] demonstrated that, by using chitosan nanoparticles (CS NPs) with the NO donor *S*-nitroso-mercaptosuccinic acid (*S*-nitroso-MSA) to provide a stable NO release, salt-stressed maize plants showed better results in improving photosynthesis, chlorophyll content, and growth, and similar results were found in soybean plants under drought conditions. Carmo et al. [122] showed that these three aspects benefit in drought conditions.

NO is vital in signaling mechanisms that help plants cope with stresses [123]. Using NO delivery systems can help plants to better thrive in adverse conditions [124,125]. Nanoparticles, particularly those made from biodegradable materials like chitosan, offer a promising solution by providing stable and controlled release of NO donors. These nano-formulations can enhance the bioactivity of NO in plants, improving stress tolerance at lower dosages. In summary, integrating NO donors with nanotechnology represents a significant advancement for sustainable agriculture, offering a robust tool to mitigate the impacts of climate change on plant systems.

## 6. Challenges and Future Directions

Despite the promising potential of NO donors and their delivery via nanoparticles, several challenges must be addressed to fully realize their antimicrobial and anti-biofilm capabilities. One of the primary tasks is the precise control of NO release kinetics. Achieving a controlled and sustained NO release at therapeutic concentrations is crucial to maximize its antimicrobial efficacy while minimizing potential cytotoxic effects on host tissues. Therefore, the development of nanoparticle designs can fine-tune the release profiles of NO donors.

Another significant issue is the potential toxicity of NO and its reactive nitrogen species. While localized delivery via nanoparticles can reduce systemic toxicity, the long-term effects of NO and RNS exposure on host cells and tissues need thorough investigation. Ensuring the biocompatibility and safety of NO-releasing nanoparticles is essential for their clinical application.

The complexity of biofilm structures poses an additional challenge. Biofilms are highly resilient and can exhibit heterogeneous responses to antimicrobial agents. Therefore, NO-releasing nanoparticles must be engineered to effectively penetrate biofilms and deliver NO uniformly across the biofilm matrix. Understanding the interactions between nanoparticles, NO, and the biofilm microenvironment is crucial for optimizing their anti-biofilm efficacy.

Another area of concern is the potential development of resistance to NO. However, NO acts through multiple mechanisms, which reduces the likelihood of resistance development. Continuous exposure to sub-lethal concentrations could potentially lead to adaptive responses in fungal and bacterial populations. Studying the long-term impact of NO treatment on bacterial populations and devising strategies to prevent resistance development are essential research areas.

Multidisciplinary research will be critical to overcoming these difficulties and fully realizing the potential of NO-releasing nanoparticles. Future research should also explore the synergistic effects of combining NO-releasing nanoparticles with other biocompatible antimicrobial agents. Such combination therapies could enhance the overall efficacy and reduce the probability of generating resistance. On the other hand, functionalizing nanoparticles with targeting ligands could improve the specificity and efficiency of NO delivery, further enhancing their therapeutic potential through multi-resistant bacteria and biofilm-related infections.

Clinical translation of NO-releasing nanoparticles involves rigorous preclinical and clinical testing to establish their safety, efficacy, and regulatory compliance. Developing standardized protocols for synthesizing, characterizing, and testing these nanoparticles will facilitate their progression from laboratory research to clinical application.

In conclusion, although NO donors and nanoparticle delivery systems show great potential for antimicrobial and anti-biofilm uses, addressing challenges related to controlled release, toxicity, biofilm penetration, resistance, and clinical application is crucial. Multidisciplinary research and collaboration will be critical to overcome these difficulties and fully realize the potential of NO-releasing nanoparticles in treating infections caused by multi-resistant bacteria and biofilm-related infections.

## Figures and Tables

**Figure 1 antibiotics-13-01047-f001:**
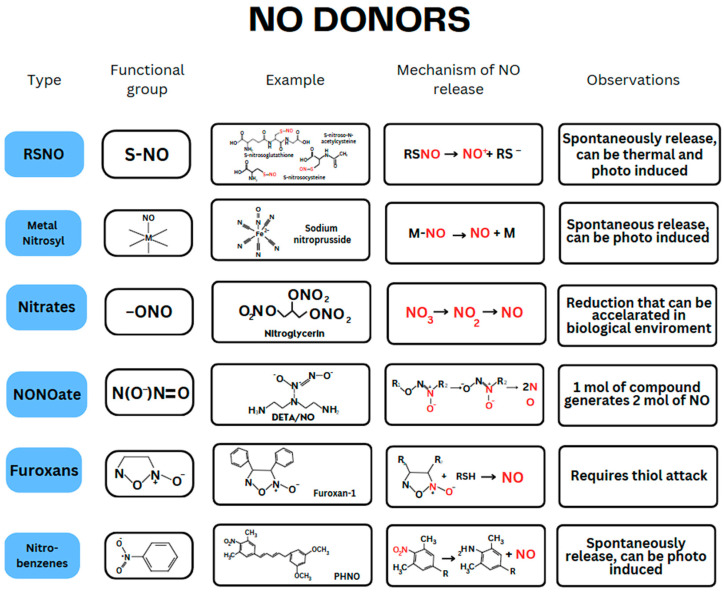
Most commonly used NO (marked in red) donors and the mechanism of NO release.

**Figure 2 antibiotics-13-01047-f002:**
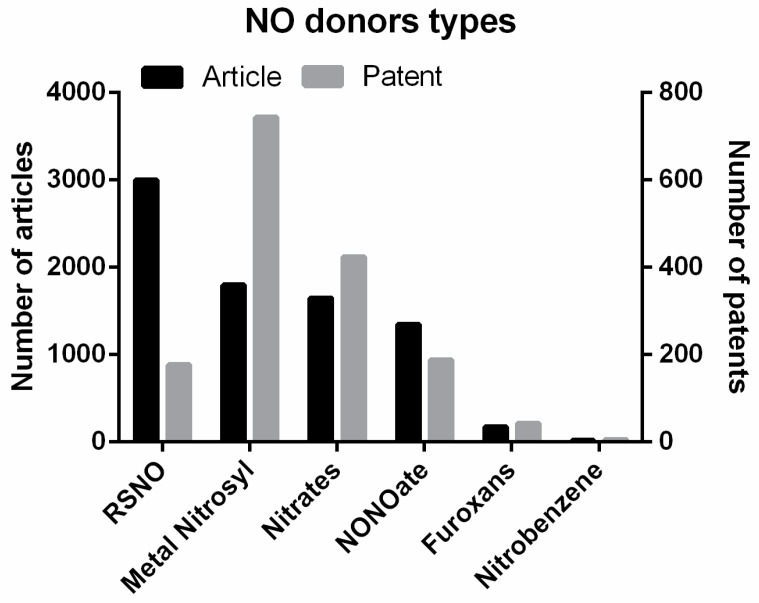
According to the Web of Science Core Collection and Derwent databases, the number of articles and patents for each type of NO donor is high. The research was conducted on 24 April 2024.

**Figure 3 antibiotics-13-01047-f003:**
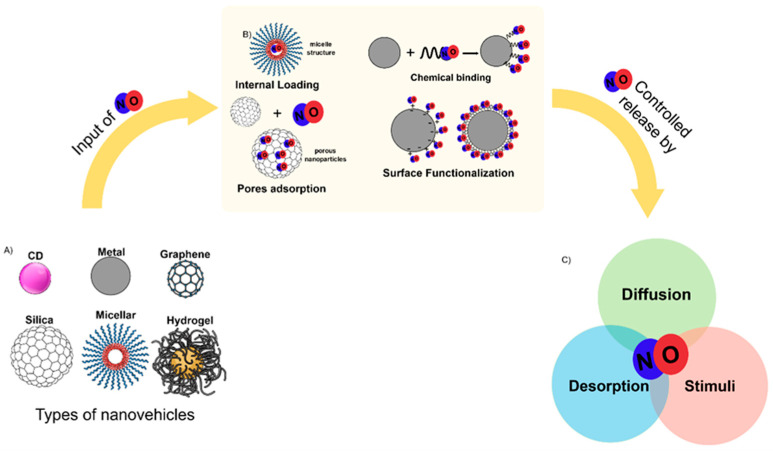
Sequence for nanoparticles with NO donors. (**A**) Type of nanovehicles studied for antimicrobial and anti-biofilm potential. DC: carbon dots. (**B**) Input of NO donors in the nanoparticles; adapted from [31]. Copyright 2023 American Chemical Society. (**C**) Types of controls for release of NO.

**Figure 4 antibiotics-13-01047-f004:**
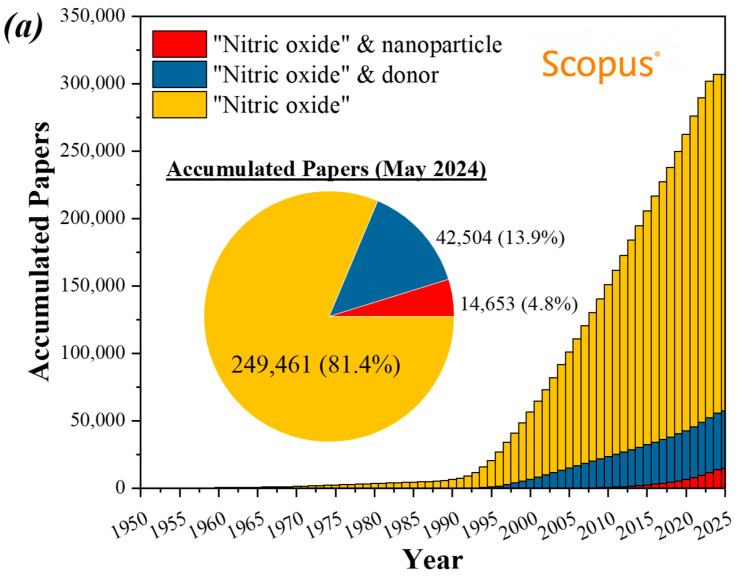
Accumulated (**a**) published papers and (**b**) granted patents over the years using “nitric oxide”, “donors”, and “nanoparticles” in the Scopus and Lens.org databases.

**Figure 5 antibiotics-13-01047-f005:**
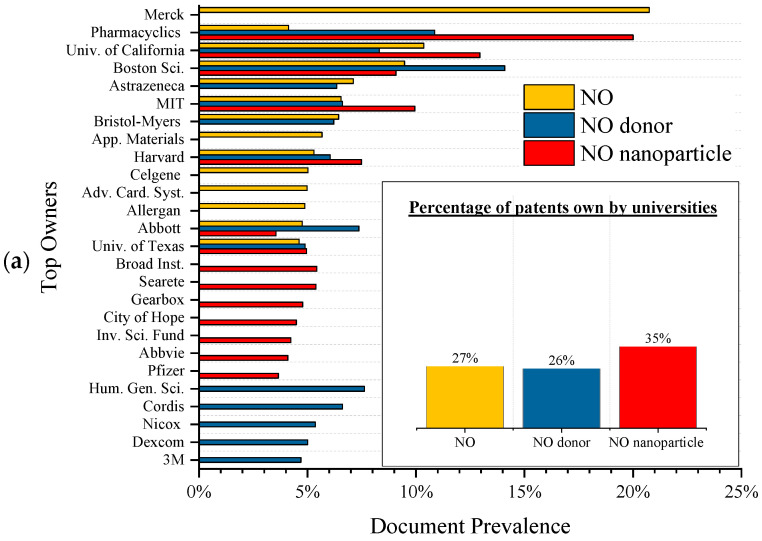
Top (**a**) owners and (**b**) CPC classification codes for the intellectual property documents in Lens.org.

## Data Availability

No new data were created or analyzed in this study.

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
