# Peer review of "Nanoparticle-Based Nitric Oxide Donors: Exploring Their Antimicrobial and Anti-Biofilm Capabilities"

_antibiotics, 2024, doi:10.3390/antibiotics13111047_

Round 1
Reviewer 1 Report
Comments and Suggestions for Authors
Summary: The review discusses nitric oxide donors and their controlled delivery by nanoparticles, for antimicrobial and anti-biofilm applications.
This review is on a timely subject likely to appeal to the Antibiotics audience. The scope of work covered is appropriate. However, the manuscript should be revised to reach a higher level of accuracy, precision, and clarity before it can be acceptable for publication. One recurring issue is that the manuscript uses qualitative terms in places where quantitative descriptions would be more appropriate and informative. Furthermore, several spelling mistakes and typos are visible throughout the manuscript.
A partial list follows below, but it is suggested that the authors take a critical approach to revising and proofreading their manuscript with this feedback in mind.
Recommendation: Reconsider after major revisions
Comments:
· Page 1, line 19-20: “However, owing to its high reactivity and short half-life…”. Please provide specific numerical values for the characteristics like high reactivity and short half-life.
· Page 1, line 25: “This synthesis underscores the potential of NO-based nanotechnologies…”. What is the synthesis the author wants to highlight in the abstract? Please rewrite the sentence for better clarity to the reader.
The manuscript discusses the NO donors and their delivery by nanoparticles; however, it does not discuss chemical synthesis.
· Page 2, line 50: “NO has gained special attention due to its antimicrobial activity against many pathogens”. Please specify in the manuscript what are these different pathogens?
· Page 2, line 51: “limited by its high reactivity and short half-life”. Please provide specific numerical values for the characteristics like high reactivity and short half-life.
· Page 2, line 61-62: “The unique physicochemical properties of nanoparticles enable them…”. What are these unique physiochemical properties that enable NO delivery? Please elaborate for clarity of reader.
· Page 3, line 95-96: “ it is not only the amount of drug at the target site, in traditional small molecule drugs, but the flux NO for a period at the target site.” Please reword and rewrite this sentence for better clarity to the reader. It seems the author wants to highlight the importance of the rate of NO release on the target site, however, the sentence does not depict this.
· Page 3, line 97: “Traditional drug strategies are ineffective in this challenge.” What is the challenge the author is referring to? Also, why are traditional drugs ineffective?
If modulating the rate of drug release is the challenge, there are literature reports on controlled release of traditional drugs.
· Page 3, line 99: “molecules capable of releasing NO…” Please replace this with NO donors.
· Page 3: Caption is missing for Figure 1. Please add an appropriate caption below Figure 1.
· Page 3, line 103: “NO donors are S-nitrosothiols (RSNO), N-diazeniumdiolate…”. Please italicize “S” and “N” respectively before the chemical names, e.g., S-nitrosothiols (RSNO) and N-diazeniumdiolate. Please do similar changes throughout the manuscript.
· Page 3, line 104: Please change “nitrobenzes” to “nitrobenzenes”. Please thorughly proofread the manuscript to rectify similar spelling mistakes.
· Page 3, Figure 1: An additional -OH functional group is drawn in the chemical structure of S-nitrosoglutathione. Please redraw the structure to rectify it.
· Page 3, Figure 1: Please change “S-nitrosoglutatione” to “S-nitrosoglutathione”. Please proofread the manuscript to rectify similar spelling mistakes.
· Page 3, Figure 1: Please change “can be photo induces” to “can be photo induced” for the observation section of Nitrobenzenes.
· Page 3, Figure 1: Please correct the spelling of “methal nitrosyl” to “Metal nitrosyl”. Please proofread the manuscript thoroughly to rectify the spelling mistakes.
· Page 3, line 137: “RSNOs are the most commonly used metal nitrosyl complexes.” Please rewrite the sentence for better clarity, as both RSNOs and metal nitrosyl complexes are separate types of NO donors.
· Page 3, line 140: “S-nitrosocysteine (CySNO), and S-nitroso-N-acetylcysteine (SNAC), as shown in Figure 1…”. The structures for CySNO and SNAC are not present in Figure 1, please include them.
· Page 4, line 143-144: “The number of commercially available RSNOs and their well-established roles and potential applications likely contribute to this disparity.” Can the author please elaborate on how it contributes to the disparity?
· Page 4, line 175-177: “acute lung injury (NCT01619280), thyroidal function (NCT00636584), and schizophrenia (NCT01548612).” Please add references for the clinical studies mentioned.
· Page 4, line 181-182: “Several clinical trials have investigated the other indications in the cardiovascular field.” Can the author provide references for the cardiovascular clinical trials mentioned, for the interested reader.
· Page 4, line 188: “ tailored by combining them with other innovative strategies, such as nanomaterials”. The author is using the keyword “nanoparticles” and ”nanomaterials” interchangeably throughout the manuscript. Please use only one of them for better clarity to the reader and avoid ambiguity.
· Page 5, line 194: “To successfully apply NO, the amount released per time is a crucial parameter to be controlled.” What application of NO is the author referring to? Please rewrite the sentence for better clarity.
· Page 5, line 201: The author mentions “non-spontaneous” in line 194 as a category of NO release mechanism. However, the author mentions “non-response” as a category in line 201. Do “non- spontaneous” and “non-response“ refer to the same NO release mechanism? If yes, please use “non- spontaneous” to maintain uniformity and for clarity to the reader.
· Page 5, line 219: Correct the typo mistake by changing ”1,5 h” to “1.5 h”.
· Page 5. Line 219: “the efficiency of S-150 was 68.4% while SNP was only 219 29.2%”. Please elaborate on what efficiency are these values depicting.
Is this efficiency of antimicrobial activity or efficiency of spontaneous release?
· Page 5, line 228: “They showed a NO release of 0.4 to 0.6 µmol of NO”.
Can the author provide the concentration ((µmol/L)), time needed for the release of 0.4 to 0.6 µmol, half-life, and the initial concentration (µmol/L) of the molecule used for the study? This information can be valuable to the reader.
· Page 7, line 281: Please change “NO precursors as isosorbide mononitrate” to “ NO precursors such as isosorbide mononitrate”.
· Page 7, line 284: What does the acronym MSNs correspond to? Please specify in line 283 that MSNs is an acronym for Mesoporous silica nanoparticles.
· Page 8, line 295: Please specify what PDA is used as an acronym for.
· Page 8, line 299: Please correct typo mistakes. Change “S.s aureus” to “S. aureus”.
· References:
Please correct the format for references 8, 77-93.
Please add the page numbers for references 16, 25, 27, 32, 34, 53, 60, 61.
Comments on the Quality of English LanguageThe quality of English is not uniform throughout the manuscript. The manuscript can benefit from thorough proofreading by a native English speaker.
Some sections could be rewritten to convey the information. Specifics are provided in the peer review comments.
Author Response
Comment 1: Page 1, line 19-20: "However, owing to its high reactivity and short half-life…". Please provide specific numerical values for the characteristics like high reactivity and short half-life.
Response 1: This paragraph was modified by "However, owing to its high reactivity due to the possession of a free radical and short half-life (1-5 seconds)". Pag 1, lines 19-20
Comment 2: Page 1, line 25: "This synthesis underscores the potential of NO-based nanotechnologies…". What is the synthesis the author wants to highlight in the abstract? Please rewrite the sentence for better clarity to the reader.
Response 2: The sentence was changed to "This review highlights the promising role of NO-based nanotechnologies in overcoming the challenges posed by increasing antibiotic resistance and bioflm-associated infections". Pag 1, lines 26-28
Comment 3: The manuscript discusses the NO donors and their delivery by nanoparticles; however, it does not discuss chemical synthesis
Response 3: Thank you very much for your comment. However, due to the large extension of the topic (synthesis and characterization of nanoparticles) this was not considered. But we added in point 3.1, the following paragraph in the text. "Numerous types of nanoparticles can serve as carriers for NO donors, each with unique characteristics and advantages. However, this review does not focus on these nanoparticle platforms' synthesis or detailed characterization. For readers interested in this aspect, we recommend consulting other comprehensive reviews on the topic, such as Rao et al. (2021), Bhardwaj and Jangde (2023), and Burlec et al. (2023). Pag 6, lines 271-274
Comment 4: Page 2, line 50: "NO has gained special attention due to its antimicrobial activity against many pathogens". Please specify in the manuscript what are these different pathogens?
Response 4: Thank you very much for your comment. The paragraph was modified by "Secondly, NO has gained special attention due to its antimicrobial activity against many pathogens, such as Pseudomonas aeruginosa, Escherichia coli, Staphylococcus aureus, Staphylococcus epidermidis, Candida albicans, Bacillus cereus, methicillin-resistant S. aureus among others" Pag 2, lines 66-69
Comment 5: Page 2, line 51: "limited by its high reactivity and short half-life". Please provide specific numerical values for the characteristics like high reactivity and short half-life
Response 5: Thank you very much for your comment. The paragraph was modified "However, the practical application of NO in clinical settings has been limited by its high reactivity and short half-life. For example, NO's reactivity is highlighted by its rapid interaction with superoxide (O2•−),3which is extremely fast and first-order concerning [O2•−]. NO has a short half-life of 1-5 sec., depending on local conditions, which contributes to its limited stability in biological systems (Lancaster et al., 2015). ." Pag 2, lines 69-73
Comment 6: Page 3, line 95-96: "it is not only the amount of drug at the target site, in traditional small molecule drugs, but the flux NO for a period at the target site." Please reword and rewrite this sentence for better clarity to the reader. It seems the author wants to highlight the importance of the rate of NO release on the target site, however, the sentence does not depict this.
Response 6: The paragraph was modified: "Thus, it is not just the concentration of the drug at the target site, as with traditional small molecule drugs, but rather the sustained rate of NO release at the target site that is crucial." Pag 3, lines 117-119.
Comment 7: Page 3, line 97: "Traditional drug strategies are ineffective in this challenge." What is the challenge the author is referring to? Also, why are traditional drugs ineffective?
Response 7: Traditional drug delivery strategies are not designed to meet this requirement effectively."
Comment 8: If modulating the rate of drug release is the challenge, there are literature reports on controlled release of traditional drugs.
Response 8: Thank you for your comment. While it is true that controlled release systems exist for traditional drugs, the challenge with nitric oxide (NO) is more complex. NO’s short half-life, high reactivity, and gaseous nature make it difficult to maintain stability and achieve sustained delivery at the target site. Traditional controlled release systems are often designed for solid or liquid drugs and may not be sufficient to meet the specific requirements of NO release. Therefore, innovative platforms, such as NO-releasing nanoparticles, have been developed to stabilize and release NO more controlled and continuously. These specialized systems address the unique challenges associated with NO delivery, which go beyond those of conventional drugs
Comment 9: Page 3, line 99: "molecules capable of releasing NO…" Please replace this with NO donors
Response 9: molecules capable of releasing NO…” was replaced by “NO donors”. Pag 3, line 121
Comment 10: Page 3: Caption is missing for Figure 1. Please add an appropriate caption below Figure 1.
Response 10: A caption for Figure 1 was added.
Comment 11: Page 3, line 103: "NO donors are S-nitrosothiols (RSNO), N-diazeniumdiolate…". Please italicize "S" and "N" respectively before the chemical names, e.g., S-nitrosothiols (RSNO) and N-diazeniumdiolate. Please do similar changes throughout the manuscript.
Response 11: This mistake was revised and corrected in the manuscript
Comment 12: Page 3, line 104: Please change "nitrobenzes" to "nitrobenzenes". Please thorughly proofread the manuscript to rectify similar spelling mistakes.
Response 12: "nitrobenzes" was changed to "nitrobenzenes".
Comment 13: Page 3, Figure 1: An additional -OH functional group is drawn in the chemical structure of S-nitrosoglutathione. Please redraw the structure to rectify it.
Response 13: The figure was enhanced according to the reviewer comments
Comment 14: Page 3, Figure 1: Please change "S-nitrosoglutatione" to "S-nitrosoglutathione". Please proofread the manuscript to rectify similar spelling mistakes
Response 14: The figure was enhanced according to the reviewer comments
Comment 15: Page 3, Figure 1: Please change "can be photo induces" to "can be photo induced" for the observation section of Nitrobenzenes
Response 15: The figure was enhanced according to the reviewer comments
Comment 16: Page 3, Figure 1: Please correct the spelling of "methal nitrosyl" to "Metal nitrosyl". Please proofread the manuscript thoroughly to rectify the spelling mistakes.
Response 16: The figure was enhanced according to the reviewer comments
Comment 17: Page 3, line 137: "RSNOs are the most commonly used metal nitrosyl complexes." Please rewrite the sentence for better clarity, as both RSNOs and metal nitrosyl complexes are separate types of NO donors.
Response 17: Thank you for the comment. The paragraph has a mistake. This was changed to "According to articles published, the RSNOs are the most used, and metal nitrosyl complexes, according to a patent registered. Pag 3, lines 133-134
Comment 18: Page 3, line 140: "S-nitrosocysteine (CySNO), and S-nitroso-N-acetylcysteine (SNAC), as shown in Figure 1…". The structures for CySNO and SNAC are not present in Figure 1, please include them.
Response 18: The figure was enhanced according to the reviewer comments
Comment 19: Page 4, line 143-144: "The number of commercially available RSNOs and their well-established roles and potential applications likely contribute to this disparity." Can the author please elaborate on how it contributes to the disparity?
Response 19: The paragraph was modified to "Figure 2 demonstrates the disproportionate ratio between patents and scientific publications. This disparity is likely influenced by the commercial availability and established roles of different NO donors, such as RSNOs and nitrates, often used in various applications, including agriculture and the environment [27, 28]. While RSNOs have more research articles, nitrates are more balanced in both patents and publications, reflecting their broader regulatory approval and commercialization." Pag 4, lines 164-169
Comment 20: Page 4, line 175-177: "acute lung injury (NCT01619280), thyroidal function (NCT00636584), and schizophrenia (NCT01548612)." Please add references for the clinical studies mentioned.
Response 20: References were added in the literature cited:
Mount Sinai Hospital, Canada. Safety Study of Nebulized Sodium Nitroprusside in Adult Acute Lung Injury. ClinicalTrials.gov 2012, NCT01619280. Available online: https://clinicaltrials.gov/ct2/show/NCT01619280 (accessed 23/10/2024).
Kavaklıdere Umut Hospital. Clinical Study Investigating the Effect of Sodium Nitroprusside Infusion on Thyroidal Function. ClinicalTrials.gov 2008, NCT00636584. Available online: https://clinicaltrials.gov/ct2/show/NCT00636584 (accessed 23/10/2024).
University of Sao Paulo. Add-on Sodium Nitroprusside to Treatment as Usual in Schizophrenia. ClinicalTrials.gov 2012, NCT01548612. Available online: https://clinicaltrials.gov/ct2/show/NCT01548612 (accessed 23/10/2024).
Comment 21: Page 4, line 181-182: "Several clinical trials have investigated the other indications in the cardiovascular field." Can the author provide references for the cardiovascular clinical trials mentioned, for the interested reader.
Response 21 : Thank you for your comment .The paragraph was rewritten and changed to "In addition, multiple clinical trials have explored other potential cardiovascular applications of nitrates, such as hypertension, treating heart failure, and preventing myocardial infarction. [39-41]" Page 5, lines 201-203.
Comment 22: Page 4, line 188: "tailored by combining them with other innovative strategies, such as nanomaterials". The author is using the keyword "nanoparticles" and" nanomaterials" interchangeably throughout the manuscript. Please use only one of them for better clarity to the reader and avoid ambiguity
Response 22: Thank you for your comment. The manuscript was homogenized using "nanoparticles". Both terms were used only in one paragraph because it was necessary to highlight their differences.
Comment 23: Page 5, line 194: "To successfully apply NO, the amount released per time is a crucial parameter to be controlled." What application of NO is the author referring to? Please rewrite the sentence for better clarity.
Response 23: The paragraph was rewritten and changed to "To achieve effective NO-based therapy, it is essential to control the rate of NO release over time, as it plays a crucial role in its therapeutic application." Page 5, lines 215-218
Comment 24: Page 5, line 201: The author mentions "non-spontaneous" in line 194 as a category of NO release mechanism. However, the author mentions "non-response" as a category in line 201. Do "non- spontaneous" and "non-response "refer to the same NO release mechanism? If yes, please use "non- spontaneous" to maintain uniformity and for clarity to the reader
Response 24: The reviewer comment is correct and was enhanced in the manuscript and replaced "non-response "by "non-spontaneous".
Comment 25: Page 5, line 219: Correct the typo mistake by changing" 1,5 h" to "1.5 h".
Response 25: This was corrected in the manuscript.
Comment 26: Page 5. Line 219: "the efficiency of S-150 was 68.4% while SNP was only 219 29.2%". Please elaborate on what efficiency are these values depicting
Response 26: Thank you very much for your comment. The efficiency is related to the reduction of biofilm biomass. Therefore, the paragraph was rewritten in the corrected manuscript. "Interestingly, within 1.5 h, the efficiency of S-150 was 68.4%, while SNP was only 29.2% in the P. aeruginosa biofilms biomass reduction." Page 6, lines 241-242
Comment 27: Is this efficiency of antimicrobial activity or efficiency of spontaneous release?
Response 27: Efficiency is related to antimicrobial activity. This was specified in the text.
Comment 28: Page 5, line 228: "They showed a NO release of 0.4 to 0.6 µmol of NO". Can the author provide the concentration ((µmol/L)), time needed for the release of 0.4 to 0.6 µmol, half-life, and the initial concentration (µmol/L) of the molecule used for the study? This information can be valuable to the reader.
Response 28: Thank you very much for your comment. The authors reported release in µmol/mg of gaseous NO in the original manuscript. The initial concentration was in mg, and no half-life was reported. The paragraph was enhanced according to the comments of the reviewer "Zhang et al. [43] synthesized a polymer based on propylene oxide with N-diazeniumdiolate moieties. They showed a NO release of 0.4, 0.5, and 0.6 µmol of gas-eous NO from 1.0 mg of the NO donor based on PEI600, PEI1800, and PEI10K. The in vitro data showed biocompatible levels, and the in vivo study showed that the groups treated with NO had higher rates of healthy tissue than infarction in rats with a model of middle cerebral artery occlusion (MCAO)" Page 6, lines 249-254.
Comment 29: Page 7, line 281: Please change "NO precursors as isosorbide mononitrate" to "NO precursors such as isosorbide mononitrate".
Response 29: This paragraph was modified according to the reviewer. Pag 8, line 308
Comment 30: Page 7, line 284: What does the acronym MSNs correspond to? Please specify in line 283 that MSNs is an acronym for Mesoporous silica nanoparticles
Response 30: The paragraph was enhanced. The final paragraph is "Mesoporous silica nanoparticles (MSNs) feature a highly ordered porous structure, providing a high surface area for loading NO donors. The tunable pore size and surface chemistry of MSNs enable controlled and targeted NO delivery, making them suitable for therapeutic applications." Pag 8, line 310-313.
Comment 31: Page 8, line 295: Please specify what PDA is used as an acronym for.
Response 31: PDA is used as an acronym for Polydopamin. The paragraph was modified by "One example is polydopamine-coated iron oxide nanoparticles (PDA-IONPs)" Pag 8, line 322-323
Comment 32: Page 8, line 299: Please correct typo mistakes. Change “S.s aureus” to “S. aureus”.
Response 32: The typo mistake was corrected.
Comment 33: Please correct the format for references 8, 77-93
Response 33: The format of the references was corrected.
Comment 34: Please add the page numbers for references 16, 25, 27, 32, 34, 53, 60, 61.
Response 34: The page numbers were added for the references solicited.

Reviewer 2 Report
Comments and Suggestions for Authors
Nanoparticle-Based Nitric Oxide Donors: Exploring Their Antimicrobial and Anti-Biofilm Capabilities
The potential of Nitric oxide (NO) an antimicrobial and anti-biofilm agent for combating biofilm-associated infections and antibiotic resistance is reviewed in this manuscript.
Following comments can be addressed before the manuscript can be accepted.
1. Discuss the health impacts of NO in the introduction section.
2. Also include the effect of other toxic compounds produced along with NO.
3. The neutralisation methods after NO production can be added, if required.
Other minor comments:
Grammatical errors:
1. Line 218, Italicise P. aeruginosa
2. Line 219, 1.5 h and not 1,5 h
3. In line 277, “used in NO delivery”
4. In line 299, “S. aureus”
5. In line 432, correct “Fe3O4@PDA@PAMAM@NONOate”
6. In line 495, remove “(”
7. In 498, correct “for resistant”
8. In line 517, change the italicised “MRSA”
9. In line 537, italicise “S. aureus”
10. In line 570, “100’s”
11. In line 740, correct “the”
12. Use either “Nitric Oxide” or “NO” consistently throughout the manuscript
13. Table 1. The “:”after “Effective against” is not required
14. Figure 2. Correct the Y-axis caption “Number of”
15. Figure 1. “spontaneously release” -> “spontaneous release” or ““spontaneously released”
Comments on the Quality of English LanguagePlease refer above
Author Response
Comment 1: Discuss the health impacts of NO in the introduction section.
Response 1: Thank you for your comments. The following paragraph was added in the introduction: In medicine, NO has been shown to regulate blood pressure, modulate immune responses, and facilitate wound healing, making it a focal point of therapeutic research [8]. Additionally, NO plays a key role in neurotransmission, vasodilation, and inflammation control, contributing to its potential applications in treating cardiovascular diseases, infections, and certain cancers [6,8]. Beyond these effects, NO has demonstrated antioxidant properties, which help mitigate oxidative stress, and has been explored as a neuroprotective agent in conditions like stroke [Chen et al., 2017] and neurodegenerative disorders [Tewari et al., 2021]. The molecule's broad range of physiological roles and therapeutic potential highlights its significance in acute treatments and chronic disease management, making it an essential focus in biomedical research. Pages 1-2, lines 37-46.
Comment 2: Also include the effect of other toxic compounds produced along with NO.
Response 2: Thank you for your comments. The following paragraph was added in the introduction: “On the other hand, in addition to nitric oxide (NO), the production of other reactive nitrogen species (RNS) can contribute to cellular toxicity. For example, when NO reacts with superoxide anion (O2•-), it forms peroxynitrite (ONOO-), a highly reactive and damaging compound that can induce oxidative and nitrosative stress [Hardy et al., 2018]. This stress leads to various forms of cellular damage, including lipid peroxidation, protein nitration, and DNA modifications, all of which play a role in the pathophysiology of human diseases. The interplay between NO and other toxic nitrogen compounds underscores the need for controlled delivery of NO to minimize undesired adverse effects while maximizing its therapeutic advantages”.Page 2, lines 47-54
Comment 3: The neutralisation methods after NO production can be added, if required.
Response 3: Thank you for the suggestion. However, this review focuses primarily on the delivery systems and controlled release of nitric oxide (NO), emphasizing its antimicrobial and anti-biofilm applications. Including neutralization, methods would expand the scope beyond the intended objective to explore the mechanisms of NO delivery and its therapeutic potential. While neutralization strategies are relevant in broader discussions of NO toxicity and safety, they are outside the scope of this review.
Comment 4: Line 218, Italicise P. aeruginosa
Response 4: Corrected in the manuscript
Comment 5: Line 219, 1.5 h and not 1,5 h
Response 5: Corrected in the manuscript
Comment 6: Line 277, “used in NO delivery”
Response 6: Corrected in the manuscript
Comment 7: In line 299, “S. aureus”
Response 7: Corrected in the manuscript
Comment 8: In line 432, correct “Fe3O4@PDA@PAMAM@NONOate
Response 8: Corrected in the manuscript
Comment 9: In line 495, remove “(”
Response 9: Corrected in the manuscript
Comment 10: In 498, correct “for resistant”
Response 10: Corrected in the manuscript
Comment 11: In line 537, italicise “S. aureus”
Response 11: Corrected in the manuscript
Comment 12: In line 570, “100’s”
Response 12 : Corrected in the manuscript
Comment 13: In line 740, correct “the
Response 13 : Corrected in the manuscript
Comment 14: Use either “Nitric Oxide” or “NO” consistently throughout the manuscript
Response 14: Corrected in the manuscript
Comment 15: Table 1. The “:”after “Effective against” is not required
Response 15: Corrected in the manuscript
Comment 16: Figure 2. Correct the Y-axis caption “Number of”
Response 16: Corrected in the manuscript
Comment 17: Figure 1. “spontaneously release” -> “spontaneous release” or ““spontaneously released”
Response 17: Corrected in the manuscript.

Reviewer 3 Report
Comments and Suggestions for Authors
This review focuses on the application of nitric oxide for delivery systems. the manuscript covers an interesting and relevant topic, however it requires some revisions to meet publication standards.
1. After introducing the abbreviation for the first time, it is not necessary to repeat it (e.g. Page 2, Line 67 mentioning Nitric Oxide (NO)).
2. please check and use italic style for In Vivo and In Vitro. e. g. page 5
3. Caption is missing for figure 1.
4. The image quality of figures 3 and 4 pages 10 and 14 needs improvement. Letters are not readable in the figures.
5. There are two figures labeled as Figure 3; please correct this duplication.
6. I recommend including a dedicated section for abbreviations in the manuscript.
7. The manuscript would benefit from further language editing.
Comments on the Quality of English LanguageThe manuscript needs further language editing.
Author Response
Comment 1: After introducing the abbreviation for the first time, it is not necessary to repeat it (e.g. Page
Response 1: Thank you for the comments. This was revised and corrected throughout the entire document.
Comment 2: Line 67 mentioning Nitric Oxide (NO)).
Response 2: Thank you for the comments. This was revised and corrected.
Comment 3: please check and use italic style for In Vivo and In Vitro. e. g. page 5
Response 3: Thank you for the comments. This was revised and corrected throughout the entire document
Comment 4: Caption is missing for Figure 1.
Response 4: Thank you for the comments. This was revised and corrected.
Comment 5: The image quality of figures 3 and 4 pages 10 and 14 needs improvement. Letters are not readable in the figures
Response 5: Figures 3 and 4 were improved
Comment 6: There are two figures labeled as Figure 3; please correct this duplication.
Response 6: The figures were labeled correctly.
Comment 7: I recommend including a dedicated section for abbreviations in the manuscript.
Response 7: A section for abbreviations was added in the manuscript (see point 7).
Comment 8: The manuscript would benefit from further language editing
Response 8: Languaje editin was checked

Round 2
Reviewer 1 Report
Comments and Suggestions for Authors
Thank you for resolving the comments. The manuscript after revisions, looks good in its present form for publication.
Reviewer 3 Report
Comments and Suggestions for Authors
The manuscript in the current version can be appropriate for publishing.